# Universal Lineage-Independent Markers of Multidrug Resistance in *Mycobacterium tuberculosis*

**DOI:** 10.3390/microorganisms12071340

**Published:** 2024-06-30

**Authors:** Hleliwe Hlanze, Awelani Mutshembele, Oleg N. Reva

**Affiliations:** 1Centre for Bioinformatics and Computational Biology, Department of Biochemistry, Genetics and Microbiology, University of Pretoria, Hillcrest, Lynnwood Rd, Pretoria 0002, South Africa; hh.hlanze@gmail.com; 2South African Medical Research Council, TB Platform, 1 Soutpansberg Road, Private Bag X385, Pretoria 0001, South Africa; awelani.mutshembele@mrc.ac.za

**Keywords:** antibiotic resistance, *Mycobacterium tuberculosis*, genetic polymorphism, GWAS

## Abstract

(1) Background: This study was aimed to identify universal genetic markers of multidrug resistance (MDR) in *Mycobacterium tuberculosis* (Mtb) and establish statistical associations among identified mutations to enhance understanding of MDR in Mtb and inform diagnostic and treatment development. (2) Methods: GWAS analysis and the statistical evaluation of identified polymorphic sites within protein-coding genes of Mtb were performed. Statistical associations between specific mutations and antibiotic resistance were established using attributable risk statistics. (3) Results: Sixty-four polymorphic sites were identified as universal markers of drug resistance, with forty-seven in PE/PPE regions and seventeen in functional genes. Mutations in genes such as *cyp123*, *fadE36*, *gidB*, and *ethA* showed significant associations with resistance to various antibiotics. Notably, mutations in *cyp123* at codon position 279 were linked to resistance to ten antibiotics. The study highlighted the role of PE/PPE and PE_PGRS genes in Mtb’s evolution towards a ‘mutator phenotype’. The pathways of acquisition of mutations forming the epistatic landscape of MDR were discussed. (4) Conclusions: This research identifies marker mutations across the Mtb genome associated with MDR. The findings provide new insights into the molecular basis of MDR acquisition in Mtb, aiding in the development of more effective diagnostics and treatments targeting these mutations to combat MDR tuberculosis.

## 1. Introduction

The bacterium *Mycobacterium tuberculosis* (Mtb), which causes tuberculosis (TB), is responsible for over a billion deaths [1]. TB is the world’s oldest pandemic, highlighting the pathogen’s ability to thrive, as it still poses a significant threat to human health today. The eradication of Mtb is challenged by the emergence of drug resistance (DR) [2]. Drug-susceptible tuberculosis is typically treated with a combination of first-line drugs: isoniazid (INH), rifampicin (RIF), ethambutol (EMB), and pyrazinamide (PZA) administered for 6 months [3]. Resistance to both INH and RIF is referred to as multidrug-resistant tuberculosis (MDR-TB) [4]. MDR-TB is commonly treated with fluoroquinolones (FQs) and injectable aminoglycosides (AGs).

In 2022, the World Health Organization (WHO) recommended an improved treatment for multidrug-resistant or rifampicin-resistant tuberculosis (MDR/RR TB) [5]. This involves a 6-month treatment plan using the drugs bedaquiline (BDQ), pretomanid, linezolid (LZD), and moxifloxacin (MXF) instead of the previously recommended 9- or 18-month regimen. If there is no resistance to FQ, a 9-month all-oral regimen is recommended [5]. Previously, extensively drug-resistant tuberculosis (XDR-TB) was defined as MDR-TB with additional resistance to FQ and at least one of the second-line injectable drugs [4]. Currently, XDR-TB is defined as showing a resistance to isoniazid, rifampicin, a fluoroquinolone, and at least one of the following: levofloxacin (LFX), BDQ, MXF, or LZD [6].

An additional definition recognized by the WHO is Pre-XDR: Pre-XDR refers to RR and MDR-TB with additional resistance to any FQ [6]. The circulation of MDR-TB and XDR-TB strains are threatening to make Mtb incurable. It is of the utmost importance that the pathogenesis and antibiotic resistance development of Mtb are fully understood. Mtb employs various mechanisms of DR that have allowed the pathogen to adapt to antibiotic treatment and to transmit and cause disease, resulting in resistant strains that are difficult to eradicate [7]. These mechanisms include drug efflux, impermeable cell walls, dormancy, compensatory evolution, epistasis, and others, contributing to Mtb’s success [7,8]. However, significant gaps remain in our understanding of these mechanisms and their role in the genetic basis of DR-TB [4]. To address these gaps, research into the genetic changes favored by the selection of DR and how these changes shape the evolutionary trajectory of DR in Mtb is needed [4,9].

Drug-resistance mutations found within specific loci in the Mtb genome, associated with several genes, have been reported and suggested as drug targets for the development of anti-tubercular drugs [10]. Well-known DR mutations that are suggested to confer high-level resistance are found in the genes *rpoB*, *inhA* and *katG*, *pncA*, *gyrA*, and *gyrB*, which confer resistance to RIF, INH, PZA, and FQ, respectively [11]. In addition to these genes, several mutations in other genes contribute to increased resistance to these antibiotics and can serve as indicators of the potential for high-level MDR [12,13]. These mutations enable Mtb bacteria to survive treatment with cocktails of antibiotics [12]. Mutations in these genes are often identified as stepping-stone mutations or prerequisite mutations [14,15], forming the epistatic landscape of MDR development [2,16,17,18,19]. The large scale of genomic data produced by whole-genome sequencing (WGS) studies has made it possible to identify such mutations, providing new insights into the evolutionary trajectory of this pathogen [20,21]. While antibiotic-resistance mutations in *M. tuberculosis* are generally well characterized, genetic markers of MDR remain understudied. To our knowledge, there are only a few publications reporting mutations in *katG* and *rpoB* conferring resistance to several first-line drugs such as isoniazid and rifampicin [22]. Another study reported the co-occurrence of mutations in *embB* and *pncA* in MDR-TB isolates in Tanzania [23].

Research into the process of DR acquisition also involves identifying mutations in non-drug target genes that lead to high-level MDR. These mutations occur in intrinsic DR mechanisms, such as drug efflux and cell wall permeability, ultimately resulting in drug tolerance and the subsequent acquisition of DR mutations in drug targets [9,24,25]. Sriraman et al. (2018) identified key genes that were associated with a predisposition towards the development of DR in vitro [24]. Additionally, mutations in DNA repair genes and in lipid biosynthesis pathways often lead to low-level resistance, which is followed by high-level DR due to stable mutations in drug targets [24]. Therefore, it is important to identify resistance-associated marker mutations and determine whether they are directly or indirectly associated with resistance to improve TB treatment [9].

An additional component that is important in the predisposition to high-level MDR in Mtb is the occurrence of a “mutator phenotype”. A mutator phenotype is suggested to provide a selective advantage in the presence of drug treatment, competition with other strains, or when resources are limited [9,21,25,26]. In our study, we hypothesized that evolution towards MDR starts with a rise of sub-populations within the pathogen characterized by an increased rate of mutations. These mutations are suggested to occur in mutation hotspots, such as PPE/PE genes. We used a statistical analysis approach to identify Mtb genomic polymorphisms forming the general epistatic landscape of MDR-TB, which include antibiotic-resistance mutations, compensatory mutations, and possible mutator phenotype markers suitable for the identification of MDR Mtb strains and the analysis of evolutionary pathways of MDR development.

## 2. Materials and Methods

### 2.1. Data Collection

The sources of sequence data for this study included the Cryptic Consortium project, which analyzed Mtb genomes from 16 countries to predict drug susceptibility to first-line TB drugs [27], and the Tuberculosis Antibiotic Resistance Catalogue Project (TB ARC) [28], available from the Pathosystems Resource Integration Centre PATRIC database [29], which now has been integrated into a new platform called the Bacterial and Viral Bioinformatics Resource Center (BV-BRC, https://www.bv-brc.org/, accessed on 26 June 2014). Other sources included the GMTV database [30] and other whole-genome sequences of Mtb and Mtb-complex [31] deposited by various authors in the European Nucleotide Archive (ENA) and GenBank databases. The list of accession numbers for the Mtb genomes used is shown in Appendix A. In all these studies, the sampled clinical Mtb strains were phenotypically tested for susceptibility to a range of antibiotics. The results of this testing were provided in the form of Appendix A. The Appendix A was meticulously checked to avoid duplications of genomes deposited in multiple databases. A total of 9388 Mtb sequences, accompanied by metadata, were gathered from these public repositories in VCF or FASTQ formats.

### 2.2. Read Mapping and Variant Calling

If only fastq files were available for a genome, the quality of the DNA reads was checked using Trimmomatic v. 0.39 [32]. DNA reads were mapped against the Mtb H37Rv (NC_000962.3) genome reference using Bowtie2 v. 2.4.4 [33]. Variant calling and filtering were performed using Sambamba v. 0.5.0 [34] and BCFtools v. 1.13 [35]. The following threshold parameters were used for filtering predicted polymorphic sites: SNP gap of 3 (-g3), indel gap of 10 (-G10), exclude (-e) SNPs that have <150 QUAL score, a DP less than 20, AF < 0.95, SP > 30, <5 reads mapped at that position, and reads in each direction that have <1 read in support of the variant. Mtb lineages were predicted for each strain using the program Resistance Sniffer and obtained VCFs as input [36].

### 2.3. Statistical Analysis

Statistical approaches for selecting discriminative polymorphisms suitable for distinguishing between strains that were susceptible or resistant to a given antibiotic are described in previous publications [19,36,37]. Metadata on antibiotic susceptibility panels were found for the following antibiotics: amikacin (AM), capreomycin (CM), cycloserine (CS), ethambutol (EMB), ethionamide (ETH), isoniazid (INH), kanamycin (KM), moxifloxacin (MFX), ofloxacin (OFX), para-aminosalicylic acid (PAS), prothionamide (PTO), pyrazinamide (PZA), rifampicin (RIF), and streptomycin (SM). To estimate the discriminative power of a polymorphism, Equation (1) was used:(1)$${Power}_{k}=1-\frac{A\cap B}{min\left(N_{A},N_{AB}\right)\;},$$
where *A∩B* is the number of strains in the lineages *A* and *B* sharing the same allelic state of the locus *k*, and *N_A_* and *N_B_* are sample sizes of the lineages *A* and *B*, respectively. Power values ranged from 0 to 1. Polymorphisms with discriminative power values greater than 0.6 were selected for downstream analysis.

The next step involved determining dependencies among the polymorphisms to identify possible epistatic interactions between them. Levin’s attributable risk statistic was used to determine directional dependencies among the polymorphisms to provide the order of acquisition of DR-associated mutations by Mtb strains. An increase in the likelihood of a substitution *A → a* when another mutation *B → b* has already occurred (*R_A→a|b_*) was calculated using Equation (2). Equation (3) was used to estimate the confidence interval Err*_A→a|b_*. Conversely, the dependence of the substitution *B → b* when the mutation *A → a* has already occurred (*R_B→b|a_*) and the respective confidence interval Err*_B→b|a_* were calculated using Equations (4) and (5).
(2)$$R_{A\to a|b}=\frac{F_{AB}F_{ab}-F_{aB}F_{Ab}}{\left(F_{AB}+F_{aB}\right)\left(F_{aB}+F_{ab}\right)}\;$$
(3)$${Err}_{A\to a|b}=\sqrt{\frac{F_{Ab}+{\left(F_{AB}+F_{ab}\right)\times R}_{A\to a|b}}{N\times F_{aB}}}\;$$
(4)$$R_{B\to b|a}=\frac{F_{AB}F_{ab}-F_{aB}F_{Ab}}{\left(F_{AB}+F_{Ab}\right)\left(F_{Ab}+F_{ab}\right)}$$
(5)$${Err}_{B\to b|a}=\sqrt{\frac{F_{aB}+{\left(F_{AB}+F_{ab}\right)\times R}_{B\to b|a}}{N\times F_{Ab}}}\;\;\;$$

In Equations (2)–(5), *F_xy_* represents the frequencies of allele combinations among sampled Mtb strains. An epistatic direction is deemed predicted when the values *R_A→a|b_* and *R_B→b|a_* differ significantly and their confidence intervals do not overlap. Epistatic interactions were visualized using CytoScape v. 3.10.2 [38]. The Excel file with data on the statistical evaluation of associations between mutations in protein-coding genes of *M. tuberculosis* and multidrug resistance and the Cytoscape 3.10.2 dataset with predicted epistatic interactions among global markers of drug resistance in the overall *M. tuberculosis* population are available from the Zenodo database [39].

## 3. Results

### 3.1. Data Collection and Computational Analysis

In this work, 9388 whole-genome sequences of Mtb, along with metadata on antibiotic susceptibility and resistance profiles, were selected from public resources. When variant calling files (VCFs) were not available from public sources, they were produced from fastq files by mapping them against the reference sequence, *M. tuberculosis* H37Rv (NC_000962.3). Filtered VCFs were used for Mtb lineage identification using the Resistance Sniffer program [36] and for creating a repository of polymorphic sites of the global Mtb population. The distribution of Mtb lineages among the selected strains is shown in Figure 1.

### 3.2. Universal Markers of Antibiotic Resistance

GWAS analysis was performed to identify genetic polymorphisms associated with antibiotic resistance in the global Mtb population. VCFs containing genetic polymorphisms were grouped by each antibiotic based on whether the respective strains had been tested for susceptibility to that antibiotic and were predicted as resistant or susceptible. Mtb strains not tested for the given antibiotics were excluded from the susceptible/resistant groups. Next, each polymorphic site in the selected VCFs was characterized by its association power (Equation (1)) with resistance to the given antibiotic. The association was deemed positive when the power value was greater than 0.6. Marker polymorphisms showing positive associations with more than five antibiotics were identified. Of these, a total of 64 polymorphic sites were identified as universal markers of drug resistance (Appendix A), with 47 sites found in PE/PPE regions and 17, as shown in Table 1, in functional genes.

### 3.3. Attributable Risk Interactions between Mutations Associated with Multidrug Resistance

We performed an attributable risk analysis on the mutations associated with resistance to many antibiotics to determine whether the development of a multi-DR phenotype follows a specific order of accumulation of mutations in protein-coding genes. Prerequisite mutations required to allow for further antibiotic-resistance mutations are known as positive epistatic interactions [18,19]. The resulting network of epistatic interactions is shown in Figure 2.

## 4. Discussion

MDR-TB strains evolve from antibiotic-susceptible bacteria due to the accumulation of specific mutations in antibiotic-resistance genes. While these mutations confer drug resistance to Mtb bacilli, they are associated with a fitness cost, reducing the viability, adaptability, and growth rate of the mutants [42,43,44]. Mathematical modeling showed that drug-induced overburdening of the resistant strains by an aggravated fitness cost would limit possible combinations of drug-resistance mutations and require other prerequisite or compensatory mutations [37,45,46,47]. In this study, we aimed to identify genetic markers of MDR-TB, including antibiotic-resistance, compensatory, and passenger mutations, through statistical analysis of the association of polymorphic sites with antibiotic-resistance patterns in 9388 whole-genome Mtb sequences obtained from various publicly available sources.

Here, we consider possible associations of the polymorphic genes shown in Table 1 and Appendix A, and their encoded proteins, with antibiotic resistance. Cytochrome P450 Cyp123 forms part of a large subfamily of cytochromes found in Mtb [27,28,43]. Lagutkin et al. (2022) conducted a GWAS study where Cyp123 at codon position M192A was associated with DR to STM, INH, and RIF [48]. The acyl-CoA dehydrogenase FadE36 protein was suggested to play a role in the lipid metabolism [49]. A recent study showed that this protein is crucial for pathogen survival in the host and is involved in drug resistance [50]. This finding agrees with the association of mutations in this protein with resistance to INH, PZA, RIF, and SM revealed in this study.

The gene *gidB* encodes S-adenosyl methionine (SAM)-dependent 7-methylguanosine (m7G) methyltransferase, which is needed for the methylation of the 16S rRNA, crucial for proper ribosome functionality. SM inhibits the ribosomal translational proofreading by binding to the methylated site in 16S rRNA. Thus, mutations in the *gidB* gene prevent methylation, thereby reducing binding affinity to SM [51]. This suggests that the prevention of methylation by GidB due to mutations renders Mtb resistant to SM. Our study showed additional associations of this mutation with resistance to several other antibiotics: AM, EMB, OFX, PTO, and PZA.

Mutations in the monooxygenase EthA are commonly known to confer resistance to ETH due to the reduction in its activity in transforming the prodrug into an active antibiotic [52,53]. Surprisingly, the performed statistical analysis did not reveal a strong association of mutations at the 337th codon of this gene with resistance to ETH, whereas associations between these mutations and resistance to AM, CM, EMB, OFX, PTO, PZA, and SM were statistically significant. This could be due to the specificity of ETH resistance development in different Mtb lineages, as the approach used in the current study focused on lineage-independent MDR development pathways. It should be noted that although the mutation at the 337th codon is mentioned in the WHO-reported list of Mtb drug-resistance mutations [40], ETH resistance was specifically associated with the mutations T88_frame-shift_, Y50C, T453I, and K448E [54,55]. The role of the genetic polymorphism at the 337th codon in MDR development requires more detailed study in the future.

MmpL12 is part of the family of MmpL transmembrane transporters. So far, no associations between MmpL mutations and DR have been reported. However, MmpL3 has been suggested to increase resistance to drugs that inhibit transporter proteins [56,57]. Additionally, the co-expression of MmpL5 and Mmps5 renders resistance to bedaquiline and clofazimine [58]. These findings suggest that the integrity of the cell wall is key to resistance to several antibiotics. Finding an association between genetic polymorphisms in membrane lipoprotein *lppP* and resistance to multiple antibiotics corroborates this hypothesis. However, there is no specific information readily available in the literature about mutations in *lppP* (Rv2330c) directly linked to antibiotic resistance.

The hydrolase Rv0045c is an esterase involved in lipid metabolism [59,60], further confirming the importance of bacterial cell wall integrity in MDR. However, no publications link this protein directly to antibiotic resistance. The current study showed that mutations at the 83rd codon were associated with resistance to AM, CM, EMB, MFX, OFX, PTO, and PZA.

An 8 bp insertion in secretory protein Rv1269c was reported by Lagutkin et al. (2022) to be associated with resistance to SM, RIF, EMB, AM, and FQ [48]. This agrees with the statistical association of mutations in this protein with multidrug resistance found in the current study. The overexpression of *opcA*, which encodes an OXPP cycle protein, was observed in INH-resistant Mtb clinical isolates [61,62]. Our analysis did not find an association between OpcA and INH resistance; however, strong statistical associations between mutations in this protein and resistance to AM, CM, MFX, OFX, and PTO were discovered.

The role of this protein in Mtb is not clear, but in cyanobacteria, it is directly involved in oxidative stress response. This may explain the overexpression of *opcA* in response to treatment with INH, as it could lower the redox potential and prevent the oxidation of the prodrug into its active form [63]. Better leveraging the cellular redox potential of Mtb cells during antibiotic treatment may explain the observed association of mutations in this protein with resistance to other antibiotics. Additionally, mutations in another protein potentially involvement in redox potential maintenance, short-chain dehydrogenase/reductase Rv1928c, showed associations with MDR. The involvement of this gene in INH resistance was reported [64]. It is likely that Rv3093c and Rv3346c are also involved in the oxidative stress response to INH exposure.

Oxidoreductase Rv3093c is part of a cluster of genes, Rv3093c–Rv3095c, suggested to be associated with ETH resistance [65]. Mutations at the codon position 207 of this protein were associated with resistance to EMB, INH, PZA, RIF, and SM. No direct associations have been reported in the literature for the conserved transmembrane protein Rv3346c, except for its up-regulation under exposure to INH in Mtb [66]. The biological significance of its association with DR is yet to be established. Resistance-associated mutations in these genes suggest the importance of redox potential maintenance in the development of MDR in Mtb.

Little is known about other genes mentioned in Table 1. Triacylglycerol synthase Rv3480c has been suggested as a biomarker to differentiate between latent and active TB [67,68]. In a study by Raman and Chandra (2008), the transcriptional regulator Rv0823c was proposed as a potential drug target [69]. According to the Mycobrowser resource (https://mycobrowser.epfl.ch; accessed on 26 June 2024), the conserved transmembrane protein Rv2434c was deemed non-essential for Mtb growth in vitro in rich media, indicating that while it may play a role in the cell, it is not critical for basic survival under these conditions. Surprisingly, our study showed that mutations at the 317th codon of this protein are associated with resistance to AM, CM, MFX, OFX, and PTO. The mechanisms of this association are not clear.

Moreover, many mutations showing strong associations with MDR were found in highly variable PE_PGRS and PE_PPE proteins (Appendix A). The role of these proteins in Mtb virulence, survival, and drug resistance is disputable. These proteins are characterized by their highly antigenic and redundant nature, allowing Mtb to modulate immune responses and evade immune-mediated clearance [70,71]. However, the genetic variability of PE_PPE and PE_PGRS proteins in Mtb seems to be more significant for their functionality than specific allelic states. These genomic loci exhibit a higher mutation density compared to the rest of the genome, with a notable proportion of these mutations being non-synonymous, which compromises the hypothesis of their functional importance but suggests their strong contribution to the overall genetic variation in Mtb populations [41].

To conclude, most of the mutations in functional genes are involved in significant cellular processes such as cell wall integrity maintenance and oxidative stress response, which are crucial for MDR development. The mycobacterial cell wall is one of many key components that enable Mtb to survive during antibiotic treatment. Mutations in genes related to cell wall integrity contribute to virulence and to the development of resistance in Mtb [72]. Another process suggested to play a role in developing high-level resistance is the leveraging of oxidative stress responses. When an individual has tuberculosis, this results in what is called an oxidative burst in the lungs [73]. This oxidative burst activates several prodrugs, such as INH [74]. The oxidative burst leads Mtb to employ counter anti-oxidative mechanisms via the accumulation of mutations in specific genes, resulting in the presence of persister cells. These persister cells can withstand TB drugs, ultimately leading to high-level resistance [73]. Amino acid substitutions in OpcA, Rv1928c, the oxidoreductase Rv3093c, and the transmembrane protein Rv3346c are likely to be part of the anti-oxidative mechanisms that Mtb employs to survive antibiotic exposure.

Statistical analysis using attributable risk statistics enabled the identification of continuous steps in the acquisition of MDR by Mtb. The ordered acquisition of antibiotic-resistance mutations can be explained by their fitness landscape [15]. MDR development may begin with mutations in different genes, followed by intermediate and compensatory mutations. The variability in the pathways of antibiotic resistance acquisition may reflect a lineage-specific distribution of antibiotic-resistance mutations [37]. Although the evolution towards DR may start in different genes, the epistatic links shown in Figure 2 demonstrate converges of this evolutionary process on a few final mutations in several PE_PGRS genes and the conserved hypothetical proteins: Rv1883c and Rv2975c. It is unclear whether these mutations are functional or were merely accumulated in highly variable genetic loci due to an increased level of mutability known as the “mutator phenotype”. Several pathways for the acquisition of mutations in functional genes can be observed in this network of epistatic links. For instance, the evolution of multidrug resistance often initiates in the monooxygenase *ethA*, the secretory protein Rv1269c, the hydrolase Rv0045c, the triacylglycerol synthase Rv3480c, and the short-chain dehydrogenase/reductase Rv1928c. The mutation in the oxidoreductase Rv3093c follows that in the latter one, whereas the former four mutations converge into a series of subsequent mutations in hypothetical and PE_PGRS genes, culminating in the mutation of the lipoprotein *lppP*.

Many mutable genes forming the epistatic network shown in Figure 2 are highly variable hypothetical and PE/PPE genes. Although the statistical analysis of associations between these genes resulted in strong links, this does not necessarily indicate the existence of specific functional connections. These genes are mutational hotspots that lead to a rise in sub-populations within the pathogen, characterized by an increased mutation rate known as the mutator phenotype. The ordering of these mutations may result from different rates of nucleotide substitutions at various genomic loci. For example, in PE_PGRS3, mutations at the 206th codon precede those at the 203rd codon (Figure 2). The increased mutation rate first identified in PE and PPE genes serves as a concerning indicator of the mutator phenotype, which results in a heightened risk of MDR development [21]. While the functional significance of PE and PPE genes is not well understood, their potential contribution to the virulence of Mtb, often linked to the ESX secretion system, should not be overlooked [75,76]. It is possible that identified initial mutations in the PE_PGRS subfamily, known to be surface-exposed cell wall proteins, affect cell wall structure and permeability or some other important functions [9,77,78].

The working hypothesis of this study assumed that MDR develops through the accumulation of mutations that either confer drug resistance or alleviate the associated fitness cost. However, other mechanisms may contribute to MDR development, such as post-translational modifications of proteins [79] or epigenetic modifications [80]. These mechanisms are not yet fully understood and need additional consideration in future studies.

## 5. Conclusions

This study was aimed to identify indicative polymorphic sites in Mtb genomes associated with MDR. The marker polymorphisms were found in genes encoding functional proteins as well as in highly variable hypothetical proteins and PE/PPE genes. While some identified polymorphisms are known from the literature as antibiotic-resistance markers, such as mutations in *cyp123*, *gidB*, *ethA*, and *lppP*, many other mutations have never been reported before. It should be noted that the detected association between genetic polymorphisms and drug resistance does not imply a direct involvement of these mutations in rendering resistance to specific antibiotics. These mutations may reflect general mechanisms of the adaptation of MDR strains to the cumulative fitness cost brought by antibiotic-resistance mutations or variations in highly mutable genomic loci due to an increased frequency of mutations in Mtb strains developing the MDR phenotype. The reported mutations should be considered MDR-TB markers rather than antibiotic-resistance mutations.

Mutations in the functional genes suggest the role of properly leveraging oxidative stress and maintaining cell integrity as key elements supporting the survival of Mtb pathogens under the pressure of multiple antibiotics. Although the identified mutations in PE_PGRS and PPE genes may not bear any functional consequences, they may serve as indicators of the mutator phenotype in Mtb isolates developing MDR.

Another aim of the study was to identify common pathways of MDR development in Mtb. Statistical analysis using attributable risk statistics revealed the continuous steps in MDR acquisition by Mtb, which were probably influenced by the fitness landscape of these mutations. However, it should be noted that this study revealed statistically significant associations between MDR and genetic polymorphisms, which have not yet been experimentally proven. Furthermore, a disparity between in vitro and in vivo antibiotic resistance is often reported, which may be due to differences in bacterial metabolism and environmental conditions within the host compared to laboratory environments.

The recently reported emergence of extensively drug-resistant (XDR) tuberculosis isolates, which are resistant to at least four of the core anti-TB drugs, or even total drug-resistant tuberculosis (TDR-TB), underlines the necessity of understanding the epistatic background underlying their evolution. This knowledge will aid in the strategic planning of measures to prevent the development and distribution of XDR and TDR-TB infections.

## Figures and Tables

**Figure 1 microorganisms-12-01340-f001:**
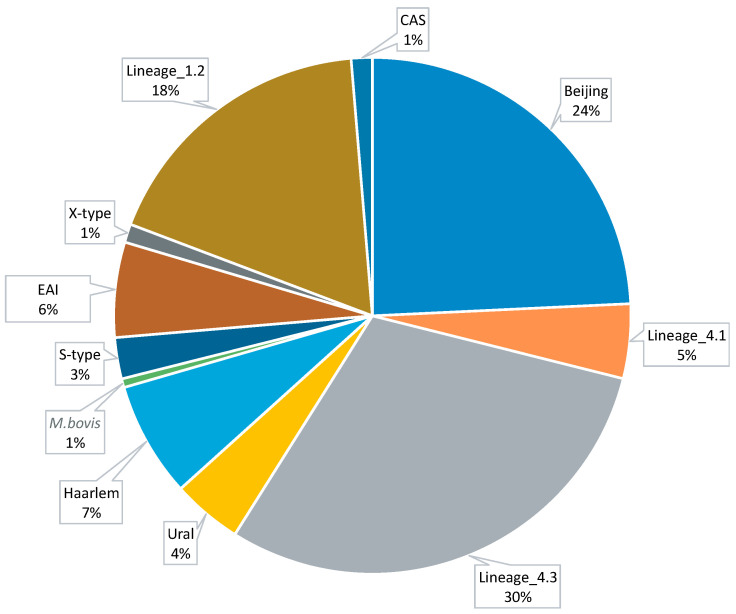
Pie diagram of the distribution of Mtb lineages among 9388 selected strains.

**Figure 2 microorganisms-12-01340-f002:**
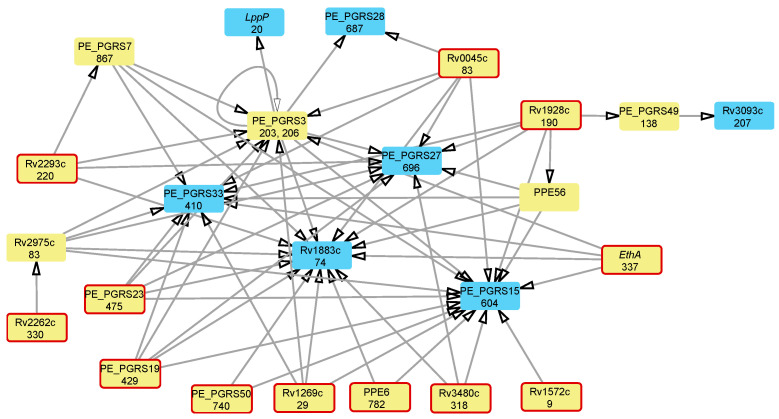
Epistatic interactions among global markers of antibiotic resistance. Nodes represent mutable genes and are labeled with the respective gene names and codon numbers where polymorphic sites are located. Epistatic links are depicted by arrows indicating the order of mutation acquisition, from prerequisite to subsequent mutations. Nodes with only outgoing epistatic links are outlined. Nodes with only incoming links are shown in an alternate blue color.

**Table 1 microorganisms-12-01340-t001:** Markers of drug resistance to more than five antibiotics found in *M. tuberculosis* genomes.

Locus Tag	Gene Product	Polymorphic Codon	Associated Antibiotic Resistance *
Rv1269c	Conserved secreted protein	29	AM, CM, **EMB**, **INH**, MFX, OFX, PTO, **PZA**, **RIF**, and SM
Rv0766c	Cytochrome P450 Cyp123	278, 279	AM, CM, **EMB**, **INH**, MFX, OFX, PTO, **PZA**, **RIF**, and SM
Rv3480c	Triacylglycerol synthase, diacylglycerol acyltransferase	318	AM, CM, **EMB**, MFX, OFX, PTO, **PZA**, and **RIF**
Rv1928c	Short-chain dehydrogenase/reductase	190	CM, **EMB**, MFX, OFX, PTO, **PZA**, and SM
Rv0045c	Possible hydrolase	83	AM, CM, **EMB**, MFX, OFX, PTO, and **PZA**
Rv3854c	Monooxygenase EthA	337 ^†^	AM, CM, **EMB**, OFX, PTO, **PZA**, and SM
Rv0823c	Transcriptional regulatory protein	322	AM, CM, **EMB**, MFX, OFX, PTO, and **PZA**
Rv3919c	Glucose-inhibited division protein GidB	65 ^†^	AM, **EMB**, OFX, PTO, **PZA**, and SM
Rv3346c	Conserved transmembrane protein	19	**EMB**, **INH**, **PZA**, **RIF**, and SM
Rv1522c	Transmembrane transport protein MmpL12	549	**EMB**, **INH**, **PZA**, **RIF**, and SM
Rv3093c	Oxidoreductase	207	**EMB**, **INH**, **PZA**, **RIF**, and SM
Rv3761c	Acyl-CoA dehydrogenase FadE36	303	**INH**, **PZA**, **RIF**, and SM
Rv2330c	Lipoprotein LppP	20	**EMB**, **INH**, **PZA**, **RIF**, and SM
Rv2434c	Conserved transmembrane protein	317	AM, CM, MFX, OFX, and PTO

* Amikacin (AM), capreomycin (CM), ethambutol (EMB), isoniazid (INH), moxifloxacin (MFX), ofloxacin (OFX), prothionamide (PTO), pyrazinamide (PZA), rifampicin (RIF), and streptomycin (SM). Antibiotics highlighted in bold represent first-line antibiotics. ^†^ Mutations at these codons were confirmed as antibiotic-resistance mutations by searching through a WHO-reported list of Mtb drug-resistance mutations [40,41].

## Data Availability

All data for this study was collected from publicly open sources.

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
