# Peer review of "Universal Lineage-Independent Markers of Multidrug Resistance in Mycobacterium tuberculosis"

_microorganisms, 2024, doi:10.3390/microorganisms12071340_

Round 1

Reviewer 1 Report

Comments and Suggestions for Authors

This paper introduces the invention of a new biomarker for identifying multidrug-resistant Mycobacterium tuberculosis. Given that M. tuberculosis is a pathogen that is difficult to eradicate in many countries due to resistance to existing first-line and second-line antibiotics, the marker system presented in this paper is both useful and valuable. The statistical analysis has been conducted appropriately, and there is no evidence of malicious manipulation of the original data, making this an honest and credible paper. The paper's development, logical structure, and references are all appropriate. However, a few essential elements need to be addressed for the paper's final publication.

1. The introduction should cover background knowledge comprehensively and in detail. Please include additional information about the current drug classes used to treat tuberculosis and their limitations.

2. Furthermore, discuss the criteria that define multidrug-resistant tuberculosis. Ensure that this discussion is based on state-of-the-art reports, rather than outdated data, to accurately reflect the current situation.

3. In the Methods section, specifically mention and briefly describe the strains of M. tuberculosis used in the study. Note that there can be differences between the strains used experimentally and those causing actual drug resistance. Describe in the Results section how you plan to overcome these discrepancies.

4. In Table 1, which lists antibiotics related to resistance, identify the most vulnerable antibiotic to resistance development. While both the authors and most readers may already know the answer, use your research findings to establish a clear connection and bridge this information.

5. In the Conclusion section, provide an outlook on how to address not only multidrug-resistant (MDR) tuberculosis but also extensively drug-resistant (XDR) tuberculosis.

Author Response

  1. The introduction should cover background knowledge comprehensively and in detail. Please include additional information about the current drug classes used to treat tuberculosis and their limitations.
  2. Furthermore, discuss the criteria that define multidrug-resistant tuberculosis. Ensure that this discussion is based on state-of-the-art reports, rather than outdated data, to accurately reflect the current situation.

Dear reviewer, thank you very much for your comments and recommendations.

To address comments 1 and 2, the Introduction was significantly modified and additional information on the antibiotics used to treat tuberculosis and multidrug resistance definitions was provided.

  1. In the Methods section, specifically mention and briefly describe the strains of M. tuberculosis used in the study. Note that there can be differences between the strains used experimentally and those causing actual drug resistance. Describe in the Results section how you plan to overcome these discrepancies.

The text in the Methods was modified; however, because public data records reported by various authors were used in this study, it was not always possible to find out whether the antibiotic resistance was observed in vivo or in vitro.

The sources of sequence data for this study included the Cryptic Consortium project, which analyzed Mtb genomes from 16 countries to predict drug susceptibility to first-line TB drugs [27], and the Tuberculosis Antibiotic Resistance Catalogue Project (TB ARC) [28], available from the Pathosystems Resource Integration Centre PATRIC database [29], which now has been integrated into a new platform called the Bacterial and Viral Bioin-formatics Resource Center (BV-BRC, https://www.bv-brc.org/). Other sources included the GMTV database [30]; and other whole genome sequences of Mtb and Mtb-complex [31] deposited by various authors in the European Nucleotide Archive (ENA) and GenBank databases. The list of accession numbers for the Mtb genomes used is shown in Suppl. Table S1. In all these studies, the sampled clinical Mtb strains were phenotypically tested for susceptibility to a range of antibiotics. The results of this testing were provided in the form of supplementary metadata. Supplementary information was meticulously checked to avoid duplications of genomes deposited in multiple databases. A total of 9,388 Mtb sequences, accompanied by metadata, were gathered from these public repositories in VCF or FASTQ formats.

  1. In Table 1, which lists antibiotics related to resistance, identify the most vulnerable antibiotic to resistance development. While both the authors and most readers may already know the answer, use your research findings to establish a clear connection and bridge this information.

First-line antibiotics were highlighted in bold in Table 1.

  1. In the Conclusion section, provide an outlook on how to address not only multidrug-resistant (MDR) tuberculosis but also extensively drug-resistant (XDR) tuberculosis.

Text was added:

However, it should be noted that this study revealed statistically significant associations between MDR and genetic polymorphisms, which have not yet been experimentally proven. Furthermore, a disparity between in vitro and in vivo antibiotic resistance is often reported, which may be due to differences in bacterial metabolism and environmental conditions within the host compared to laboratory environments.

The recently reported emergence of extensively drug-resistant (XDR) tuberculosis isolates, which are resistant to at least four of the core anti-TB drugs, or even total drug-resistant tuberculosis (TDR-TB), underlines the necessity of understanding the epi-static background underlying their evolution. This knowledge will aid in the strategic planning of measures to prevent the development and distribution of XDR and TDR-TB infections.

Reviewer 2 Report

Comments and Suggestions for Authors

The manuscript by Hlantse et al. titled "Universal lineage-independent markers of multidrug resistance in M. tuberculosis" is dedicated to the search for markers of antibiotic resistance in tuberculosis mycobacteria through a sophisticated GWAS analysis. The manuscript is quite interesting and well-written.

However, I have a few comments for improvement before publication:

- It is better to fully write out the species name Mycobacterium tuberculosis in the title.

- Lines 46-48: The correspondence of 6 genes to 4 antibiotics is not entirely clear.

- Line 92: I would like to know how many genomes were taken from which databases, as most of them are duplicates.

- Line 161: 47 and 17, but Table 1 shows 14 loci and 15 polymorphic codons. Then why the reference to this Table?

- The section from line 171 to 258, as well as the paragraph from lines 274-288, should be placed in the Discussion section.

Author Response

The manuscript by Hlantse et al. titled "Universal lineage-independent markers of multidrug resistance in M. tuberculosis" is dedicated to the search for markers of antibiotic resistance in tuberculosis mycobacteria through a sophisticated GWAS analysis. The manuscript is quite interesting and well-written.

Dear reviewer,

Thank you very much for your comments and suggestions.

However, I have a few comments for improvement before publication:

- It is better to fully write out the species name Mycobacterium tuberculosis in the title.

- fixed.

- Lines 46-48: The correspondence of 6 genes to 4 antibiotics is not entirely clear.

The text was changed: “Well-known DR mutations that are suggested to confer high-level resistance are found in the genes rpoB, inhA and katG, pncA, gyrA and gyrB, which confer resistance to rifampicin (RIF), isoniazid (INH), pyrazinamide (PZA), and fluoroquinolones (FQs) respectively [11].”

- Line 92: I would like to know how many genomes were taken from which databases, as most of them are duplicates.

The text was modified and new Suppl. Table S1 was added:

The sources of sequence data for this study included the Cryptic Consortium project, which analyzed Mtb genomes from 16 countries to predict drug susceptibility to first-line TB drugs [27], and the Tuberculosis Antibiotic Resistance Catalogue Project (TB ARC) [28], available from the Pathosystems Resource Integration Centre PATRIC database [29], which now has been integrated into a new platform called the Bacterial and Viral Bioin-formatics Resource Center (BV-BRC, https://www.bv-brc.org/). Other sources included the GMTV database [30]; and other whole genome sequences of Mtb and Mtb-complex [31] deposited by various authors in the European Nucleotide Archive (ENA) and GenBank databases. The list of accession numbers for the Mtb genomes used is shown in Suppl. Table S1. In all these studies, the sampled clinical Mtb strains were phenotypically tested for susceptibility to a range of antibiotics. The results of this testing were provided in the form of supplementary metadata. Supplementary information was meticulously checked to avoid duplications of genomes deposited in multiple databases. A total of 9,388 Mtb sequences, accompanied by metadata, were gathered from these public repositories in VCF or FASTQ formats.

Suppl. Table S1. List of accession numbers of M. tuberculosis and Mtb-complex genomes at public depositaries: ERR###### – the European Nucleotide Archive (ENA); SRR####### – GenBank; TB#### – GMTV; and m#### or mal###### – BV-BRC.

- Line 161: 47 and 17, but Table 1 shows 14 loci and 15 polymorphic codons. Then why the reference to this Table?

The text was modified: “Of these, a total of 64 polymorphic sites were identified as universal markers of drug resistance (Suppl. Table S2), with 47 sites found in PE/PPE regions and 17, shown in Table 1, in functional genes.”

- The section from line 171 to 258, as well as the paragraph from lines 274-288, should be placed in the Discussion section.

Done as requested.